# Influence of Plastic Anisotropy on the Limit Load of an Overmatched Cracked Tension Specimen

**Elena Lyamina [1,2,*]⬤, Nataliya Kalenova [3] and Dinh Kien Nguyen [4,5]**

[1] Division of Computational Mathematics and Engineering, Institute for Computational Science,
Ton Duc Thang University, 19 Nguyen Huu Tho St, Tan Phong Ward, Dist 7,
Ho Chi Minh City 700000, Vietnam

[2] Faculty of Civil Engineering, Ton Duc Thang University, 19 Nguyen Huu Tho St, Tan Phong Ward, Dist 7,
Ho Chi Minh City 700000, Vietnam

[3] Moscow Aviation Institute (National Research University), Volokolamskoe Shosse 4, 125993 Moscow, Russia;
perepljuika@bk.ru

[4] Institute of Mechanics, VAST, 18 Hoang Quoc Viet, Hanoi 100000, Vietnam; ndkien@imech.vast.vn

[5] Faculty of Mechanics and Automation, Graduate University of Science and Technology, VAST,
18 Hoang Quoc Viet, Hanoi 100000, Vietnam

\* Correspondence: lyaminaea@tdtu.edu.vn; Tel.: +84-2837755024

**Abstract:** Plastic anisotropy is a common property of many metallic materials. This property affects many aspects of structural analysis and design. In contrast to the isotropic case, there is a great variety of yield criteria proposed for anisotropic materials. Moreover, even if one specific yield criterion is selected, several constitutive parameters are involved in it. Therefore, parametric analysis of structures made of anisotropic materials is quite cumbersome. The present paper demonstrates the effect of the constitutive parameters involved in Hill's quadratic yield criterion on the upper bound limit load for weld stretched overmatched tension specimens containing a crack of arbitrary shape, assuming that the crack is located inside the weld. Different sets of the constitutive parameters are involved in the yield criteria for weld and base materials. Since the limit load is an input parameter of many flaw assessment procedures, the final result of the present paper shows that it is necessary to take into account plastic anisotropy in these procedures. It is worthy of note that the limit load is involved in the flaw assessment procedures in combination with the stress and strain fields near the tip of a crack. In anisotropic materials, these fields may become non-symmetric even under symmetric loading. This behavior affects the propagation of cracks.

**Keywords:** flaw assessment procedures; limit load; plastic anisotropy; welded structures

---

## 1. Introduction

The defect assessment procedures are widely used in engineering practice for assessing the integrity of structural components containing cracks and other defects. The basic principles of such procedures have been presented, for example, in [1]. A vital input parameter of the defect assessment procedures is the limit load. Several reviews of limit load solutions are available in the literature [2–4]. Most of these solutions are upper bound solutions for isotropic materials. However, it has been noted in [4] that plastic anisotropy has a significant effect on the limit load. Therefore, it is important to study this effect on the limit load for various components that are widely used in engineering practice. Several solutions are available in the literature. Upper bound limit loads for the standard overmatched middle cracked tension specimen and overmatched cracked plate in pure bending have been derived in [5,6], respectively. In both cases, the weld material has been assumed to be isotropic. The base material obeys the orthotropic yield criterion proposed in [7]. Three solutions are available

for highly undermatched specimens [8–10]. Paper [8] deals with the standard middle cracked tension specimen, paper [9] with the cracked plate in pure bending, and paper [10] with scarf joints containing a crack. These solutions are based on the assumption that the base material is rigid. The kinematically admissible velocity fields have been chosen to satisfy the asymptotic behavior of the real velocity field near the bi-material interfaces, which is known from the general theory of plasticity [11].

In the present paper, a welded specimen with a through crack subject to tensile loading under plane strain deformation is considered assuming that both weld and base materials are orthotropic. Several solutions for such specimens made of isotropic materials have been proposed in [12–15]. The middle cracked tension specimen is a particular case of the tension specimen under consideration. The effect of plastic anisotropy on the limit load for this specific type of tension specimen has been evaluated in [5,8]. A distinguishing feature of the solution provided in the present paper is that the crack is not straight and is arbitrarily located inside the weld. Some restrictions on the shape of the crack apply. These restrictions are specified in due course. Another solution for a tension specimen with a through crack of arbitrary shape has been provided in [16]. This solution is for a highly untermatched welded joint made of isotropic materials.

The orthotropic yield criterion proposed in [7] is adopted. Under plane strain conditions, this criterion contains two material parameters that control plastic anisotropy. These parameters are different for the weld and base materials. Besides, there are several geometric parameters. Therefore, the total number of parameters that should be specified for a given specimen is quite large. Under such conditions, the finite element method that is often used for finding limit load solutions (for example, Refs [17–19]) is not efficient. In the present paper, a semi-analytic upper bound solution based on a discontinuous velocity field is provided. The final solution involves an intermediate solution of an auxiliary problem and some algebra. The intermediate solution requires a simple numerical treatment. This solution can then be used for finding the final solution for any specimen of the type considered.

A growing body of literature has examined the effect of plastic anisotropy on various aspects of ductile fracture. The distribution of stresses and strains near a crack tip has been found numerically under plane strain conditions in [20]. The yield criterion proposed in [7] has been adopted. It has been shown that crack tip opening displacement (CTOD) is nearly unaffected by introducing the plastic anisotropy. It is worthy of note that engineering treatment model (ETM), which is a defect assessment procedure, is based on CTOD [21]. Therefore, the effect of the plastic anisotropy on the prediction of this procedure is revealed only through the limit load solution found in the present paper. Theories of ductile fracture that account for initial plastic anisotropy and microstructure evolution have been proposed in [22,23]. In [22], the initiation of a crack in notched bars has been predicted employing a new measure of ductility that incorporates plastic anisotropy. In [23], it is shown that it is impossible to obtain a unified description of rupture properties for several specimens tested along different directions without accounting for plastic anisotropy. The failure strain in tensile tests has been predicted in [24] using nonlinear finite element simulations and strain localization analyses. The constitutive equations account for plastic anisotropy, strength and work hardening. It has been found that plastic anisotropy significantly affects tensile ductility. The papers [20,22–24] employ sophisticated numerical methods for solving problems. However, analytic and semi-analytic solutions are more useful for a wide class of engineering problems in fracture mechanics [25,26].

## 2. Statement of the Problem

Welded joints subject to tensile loading are often idealized as a plane-strain specimen (see, for example, [3,13,15]). The geometry of the specimen considered in the present paper is shown in Figure 1. The Cartesian coordinate system $(x, y)$ has been introduced to describe the system of loading and the geometry of the crack. For this purpose, it is sufficient to specify the direction of the coordinate axes. The origin of the coordinate system will be chosen later. Two Cartesian coordinates determine each of the crack tips. The present study is restricted to cracks whose shape does not affect the solution. The cracks that satisfy this condition will be specified later. The direction of the tensile forces is parallel

to the $y$-axis. It is assumed that the cross-sections where the tensile forces are applied are rigid. Each of these cross-sections moves with velocity $U$, and the direction of the velocity coincides with the direction of the corresponding force.

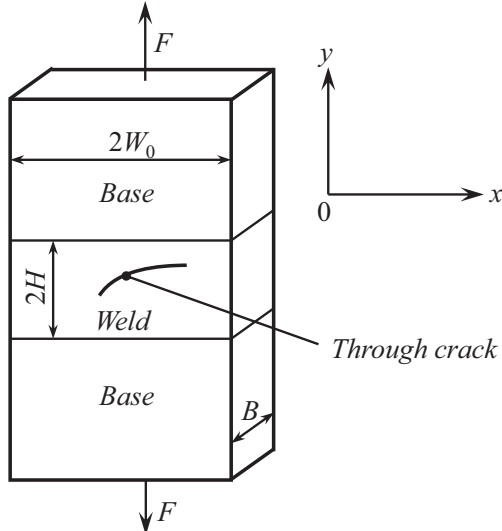

**Figure 1.** Geometry of the specimen.

Both the weld and base materials are orthotropic with the principal axes of anisotropy coinciding with $x$- and $y$-coordinate lines. This type of anisotropy in the base material is typical in sheets produced by rolling. It is assumed that the materials obey Hill's quadratic yield criterion [7]. In the case of plane strain deformation this criterion reads

$$\frac{\left(\sigma_{xx} - \sigma_{yy}\right)^2}{4\left(1 - c\right)} + \sigma_{xy}^2 = T^2 \tag{1}$$

where $\sigma_{xx}$, $\sigma_{yy}$ and $\sigma_{xy}$ are the components of the stress tensor in the Cartesian coordinates, $T$ is the shear yield stress in the $xy$ – plane and $c$ is

$$c = 1 - \frac{K + G}{4T^2\left(KG + GS + SK\right)} \tag{2}$$

The parameters involved in (2) are expressible in terms of the yield stresses in respect to the principal axes of anisotropy. In particular,

$$2K = \frac{1}{Y^2} + \frac{1}{Z^2} - \frac{1}{X^2}, \quad 2G = \frac{1}{Z^2} + \frac{1}{X^2} - \frac{1}{Y^2}, \quad 2S = \frac{1}{X^2} + \frac{1}{Y^2} - \frac{1}{Z^2} \tag{3}$$

where $X$, $Y$, and $Z$ are the tensile yield stresses in the $x$-, $y$- and thickness directions, respectively. Theoretically, the value of $c$ can vary in the interval $-\infty < c < 1$. The isotropic material is obtained at $c = 0$. Using (2) the yield criterion for the weld and base materials can be written in the form

$$\frac{\left(\sigma_{xx} - \sigma_{yy}\right)^2}{4\left(1 - c_W\right)} + \sigma_{xy}^2 = T_W^2 \quad \text{and} \quad \frac{\left(\sigma_{xx} - \sigma_{yy}\right)^2}{4\left(1 - c_B\right)} + \sigma_{xy}^2 = T_B^2 \tag{4}$$

Here, the subscript "$W$" is related to the weld and the subscript "$B$" to the base material.

### 3. Solution for an Auxiliary Problem

The configuration of the specimen under consideration and the adopted Cartesian coordinate system $(x, y)$ are shown in Figure 2a. This specimen contains no crack.

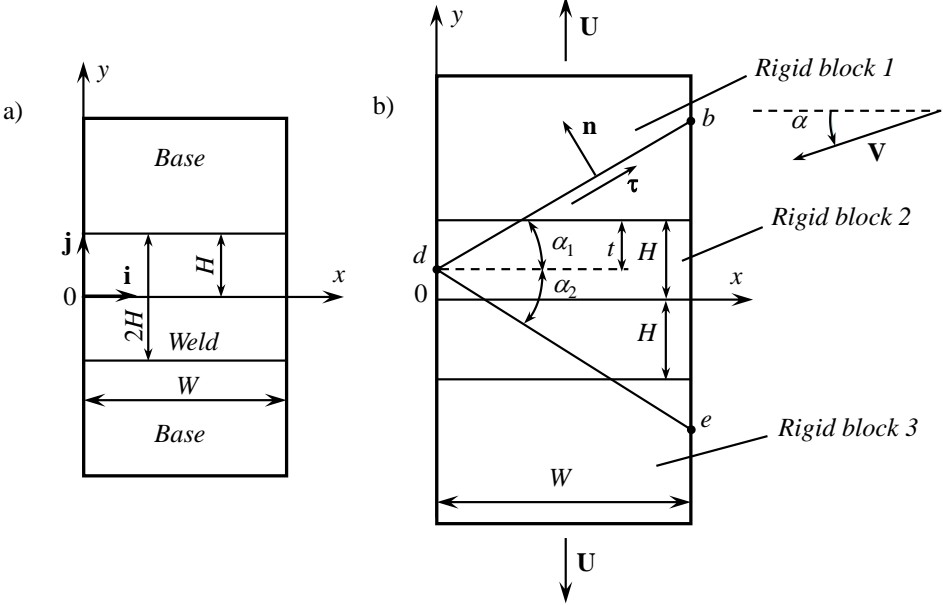

**Figure 2.** (**a**) Geometry of the auxiliary problem; (**b**) kinematically admissible velocity field.

### 3.1. Kinematically Admissible Velocity Field

Consider the kinematically admissible velocity field illustrated in Figure 2b. Rigid blocks 1 and 3 move along the *y*-axis in the opposite directions. The velocity of each block is equal to *U*. This velocity is given. Rigid block 2 moves with velocity *V*. The direction of this velocity vector is controlled by angle $\alpha$. The velocity field is discontinuous across lines *db* and *de*. The angle between the *x*-axis and line *db* is denoted as $\alpha_1$ and between the *x*-axis and line *de* as $\alpha_2$. Lines *db* and *de* intersect at point *d*. This point lies on the *y*-axis and its *y*-coordinate is determined by the value of *t*. It is assumed that the value of *t* is prescribed. The values of *V*, $\alpha$, $\alpha_1$ and $\alpha_2$ should be found from the solution.

The normal velocity across the velocity discontinuity lines must be continuous. Consider line *db*. The unit normal vector can be represented as

$$\mathbf{n} = -\sin\alpha_1 \mathbf{i} + \cos\alpha_1 \mathbf{j} \tag{5}$$

where **i** and **j** are the base vectors of the Cartesian coordinate system (Figure 2a). The velocity of rigid blocks 1 and 2 are represented as

$$\mathbf{U} = U\mathbf{j} \quad \text{and} \quad \mathbf{V} = -V\cos\alpha\mathbf{i} - V\sin\alpha\mathbf{j} \tag{6}$$

respectively. Substituting (5) and (6) into the equation $\mathbf{U} \cdot \mathbf{n} = \mathbf{V} \cdot \mathbf{n}$ one gets

$$U\cos\alpha_1 = V\sin(\alpha_1 - \alpha). \tag{7}$$

Line *de* can be treated in a similar manner. As a result,

$$U\cos\alpha_2 = V\sin(\alpha_2 + \alpha). \tag{8}$$

Solving Equations (7) and (8) for *V* and $\alpha$ one arrives at

$$V = U\frac{\cos\alpha_2}{\sin(\alpha_2 + \alpha)} \quad \text{and} \quad \tan\alpha = \frac{\sin(\alpha_1 - \alpha_2)}{2\cos\alpha_1\cos\alpha_2}. \tag{9}$$

Return to line *db*. The unit vector tangent to this line is represented as

$$\boldsymbol{\tau} = \cos \alpha_1 \mathbf{i} + \sin \alpha_1 \mathbf{j}. \tag{10}$$

The amount of velocity discontinuity across line *db* is determined from the equation

$$[u]_{db} = \mathbf{U} \cdot \boldsymbol{\tau} - \mathbf{V} \cdot \boldsymbol{\tau}. \tag{11}$$

Equations (6) and (11) combine to give

$$[u]_{db} = U \frac{\cos \alpha}{\sin (\alpha_1 - \alpha)}. \tag{12}$$

The amount of velocity discontinuity across line *de* is determined in a similar manner. As a result,

$$[u]_{de} = U \frac{\cos \alpha}{\sin (\alpha_2 + \alpha)}. \tag{13}$$

One can eliminate $\alpha$ in (12) and (13) using the second equation in (9). The value of $U$ is immaterial. The values of $\alpha_1$ and $\alpha_2$ should be found from the upper bound theorem.

### 3.2. Plastic Work Rate

It is seen from the general structure of the kinematically admissible velocity field that there is no plastic region of finite size. Therefore, one needs to find the plastic work rate at the velocity discontinuities lines only. Consider line *db*. Let $L_{db}^{(W)}$ and $L_{db}^{(B)}$ be the length of this line within the weld and base materials, respectively. Note that $L_{db}^{(B)} = 0$ if $\alpha_1 \leq \alpha_1^{(c)}$ where (Figure 2b)

$$\tan \alpha_1^{(c)} = \frac{t}{W}. \tag{14}$$

It follows from the geometry of Figure 2b that

$$L_{db}^{(W)} = \begin{cases} t/\sin \alpha_1 & \text{if } \alpha_1 > \alpha_1^{(c)} \\ W/\cos \alpha_1 & \text{if } \alpha_1 \leq \alpha_1^{(c)} \end{cases} \quad \text{and} \quad L_{db}^{(B)} = \begin{cases} \dfrac{W}{\cos \alpha_1} - \dfrac{t}{\sin \alpha_1} & \text{if } \alpha_1 > \alpha_1^{(c)} \\ 0 & \text{if } \alpha_1 \leq \alpha_1^{(c)}. \end{cases} \tag{15}$$

Then, the plastic work rate at velocity discontinuity line *db* is

$$\Omega_{db} = \left( k_{db}^{(W)} L_{db}^{(W)} + k_{db}^{(B)} L_{db}^{(B)} \right) [u]_{db} B. \tag{16}$$

Here, $B$ is the thickness of the specimen (Figure 1), $k_{db}^{(W)}$ is the shear stress at velocity discontinuity line *db* within the weld material, and $k_{db}^{(B)}$ is the shear stress at velocity discontinuity line *db* within the base material.

The plastic work rate at velocity discontinuity line *de* can be determined in a similar manner. In particular,

$$\Omega_{de} = \left( k_{de}^{(W)} L_{de}^{(W)} + k_{de}^{(B)} L_{de}^{(B)} \right) [u]_{de} B \tag{17}$$

where $k_{de}^{(W)}$ is the shear stress at velocity discontinuity line *de* within the weld material, $k_{de}^{(B)}$ is the shear stress at velocity discontinuity line *de* within the base material, $L_{de}^{(W)}$ is the length of velocity discontinuity line *de* within the weld material, and $L_{db}^{(B)}$ is the length of velocity discontinuity line *de* within the base material. It is seen from the geometry of Figure 2b that

$$L_{de}^{(W)} = \begin{cases} (2H - t)/\sin\alpha_2 & \text{if } \alpha_2 > \alpha_2^{(c)} \\ W/\cos\alpha_2 & \text{if } \alpha_2 \le \alpha_2^{(c)} \end{cases} \quad \text{and} \quad L_{de}^{(B)} = \begin{cases} \dfrac{W}{\cos\alpha_2} - \dfrac{(2H - t)}{\sin\alpha_2} & \text{if } \alpha_2 > \alpha_2^{(c)} \\ 0 & \text{if } \alpha_2 \le \alpha_2^{(c)}. \end{cases} \tag{18}$$

$$\tan\alpha_2^{(c)} = \frac{2H - t}{W}. \tag{19}$$

The shear stress at velocity discontinuity lines in the material under consideration is given by [7]

$$k = T\sqrt{1 - c\sin^2 2\theta}. \tag{20}$$

where $\theta$ is the orientation of the velocity discontinuity line relative the *x*-axis. It follows from Equation (20) and Figure 2b that

$$k_{db}^{(W)} = T_W\sqrt{1 - c_W\sin^2 2\alpha_1}, \quad k_{db}^{(B)} = T_B\sqrt{1 - c_B\sin^2 2\alpha_1},$$
$$k_{de}^{(W)} = T_W\sqrt{1 - c_W\sin^2 2\alpha_2}, \quad k_{de}^{(B)} = T_B\sqrt{1 - c_B\sin^2 2\alpha_2}. \tag{21}$$

The total work rate is determined from (16) and (17) as

$$\Omega_0 = \Omega_{db} + \Omega_{de} = \left(k_{db}^{(W)}L_{db}^{(W)} + k_{db}^{(B)}L_{db}^{(B)}\right)[u]_{db}B + \left(k_{de}^{(W)}L_{de}^{(W)} + k_{de}^{(B)}L_{de}^{(B)}\right)[u]_{de}B. \tag{22}$$

Using (12) and (13), one can transform this equation to

$$\Omega = \frac{\Omega_0}{WBUT_B} = \left(\frac{k_{db}^{(W)}}{T_B}\frac{L_{db}^{(W)}}{W} + \frac{k_{db}^{(B)}}{T_B}\frac{L_{db}^{(B)}}{W}\right)\frac{\cos\alpha}{\sin(\alpha_1 - \alpha)} + $$
$$\left(\frac{k_{de}^{(W)}}{T_B}\frac{L_{de}^{(W)}}{W} + \frac{k_{de}^{(B)}}{T_B}\frac{L_{de}^{(B)}}{W}\right)\frac{\cos\alpha}{\sin(\alpha_2 + \alpha)}. \tag{23}$$

Moreover, it follows from (21) that

$$\frac{k_{db}^{(W)}}{T_B} = M\sqrt{1 - c_W\sin^2 2\alpha_1}, \quad \frac{k_{db}^{(B)}}{T_B} = \sqrt{1 - c_B\sin^2 2\alpha_1},$$
$$\frac{k_{de}^{(W)}}{T_B} = M\sqrt{1 - c_W\sin^2 2\alpha_2}, \quad \frac{k_{de}^{(B)}}{T_B} = \sqrt{1 - c_B\sin^2 2\alpha_2}. \tag{24}$$

where $M = T_W/T_B$. One can rewrite Equations (15) and (18) as

$$\frac{L_{db}^{(W)}}{W} = \begin{cases} \dfrac{t}{W\sin\alpha_1} & \text{if } \alpha_1 > \alpha_1^{(c)} \\ 1/\cos\alpha_1 & \text{if } \alpha_1 \le \alpha_1^{(c)} \end{cases} \quad \text{and} \quad \frac{L_{db}^{(B)}}{W} = \begin{cases} \dfrac{1}{\cos\alpha_1} - \dfrac{t}{W\sin\alpha_1} & \text{if } \alpha_1 > \alpha_1^{(c)} \\ 0 & \text{if } \alpha_1 \le \alpha_1^{(c)} \end{cases} \tag{25}$$

and

$$\frac{L_{de}^{(W)}}{W} = \begin{cases} \left(\dfrac{2H}{W} - \dfrac{t}{W}\right)/\sin\alpha_2 & \text{if } \alpha_2 > \alpha_2^{(c)} \\ 1/\cos\alpha_2 & \text{if } \alpha_2 \le \alpha_2^{(c)} \end{cases} \quad \text{and} \quad \frac{L_{de}^{(B)}}{W} = \begin{cases} \dfrac{1}{\cos\alpha_2} - \left(2\dfrac{H}{W} - \dfrac{t}{W}\right)\dfrac{1}{\sin\alpha_2} & \text{if } \alpha_2 > \alpha_2^{(c)} \\ 0 & \text{if } \alpha_2 \le \alpha_2^{(c)}, \end{cases} \tag{26}$$

respectively. It is seen from (9), (14), (19), (24)–(26) that the right-hand side of (23) depends on seven independent dimensionless parameters $t/W$, $H/W$, $M$, $c_W$, $c_B$, $\alpha_1$, and $\alpha_2$. The first five are prescribed, and the last two should be found by minimizing the right-hand side of (23). The minimum value of $\Omega$ is denoted as $\Omega_{\min}$.

Since many independent parameters are involved in the final solution, its full parametric analysis and graphical illustration are not feasible. As two examples, Figures 3 and 4 show the variation of $\Omega_{\min}$ with $c_w$ at several values of $c_b$. In all cases, $M = 3$. The curves in Figure 3 correspond to $h = 0.3$ and $t = H$, and in Figure 4 to $h = 0.1$ and $t = 0$. It is seen from both figures that the effect of plastic anisotropy on $\Omega_{\min}$ (i.e., on the limit load) is very significant.

It is seen from Figures 3 and 4 that $\Omega_{\min}$ is a monotonically decreasing function of $c_b$ at a given value of $c_w$. Analogously, $\Omega_{\min}$ is a monotonically decreasing function of $c_w$ at a given value of $c_b$. Therefore, $\Omega_{\min}$ in the domain $c_b > 0$ and $c_w > 0$ is smaller than the value of $\Omega_{\min}$ for the isotropic case. It is worthy of note that this comparison between the isotropic and anisotropic materials involves the assumption that the shear yield stress of the isotropic material is equal to $T$. It follows from Equation (1) at $c = 0$. However, there is no rational reason for making this assumption. For example, it is possible to choose that the tensile yield stress of the isotropic material is equal to $X$. In this case, the shear yield stress of this material is equal to $X/\sqrt{3}$. Hence, the dimensionless quantity $\Omega_{\min}$ found from the solution should be multiplied by $X/\left(\sqrt{3}T\right)$ for comparison with the anisotropic material. Therefore, it is not reasonable to speak of the positive or negative effect of plastic anisotropy on the limit load without having the real properties of isotropic and anisotropic materials. For, the choice of the isotropic reference material affects the result of the comparison between the theoretical solutions. On the other hand, it is impossible to make this choice unambiguously.

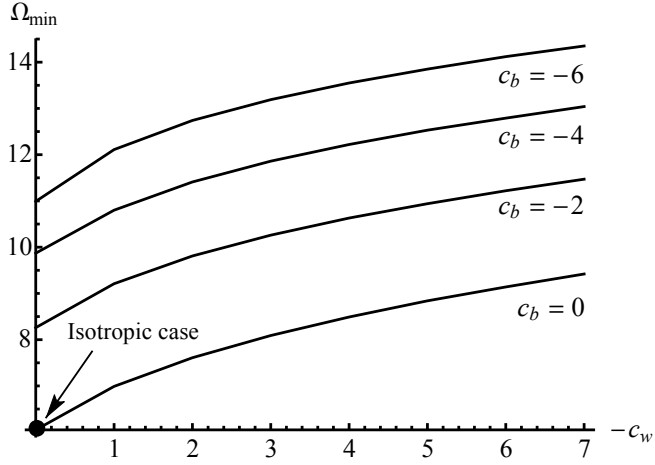

**Figure 3.** Effect of plastic anisotropy on $\Omega_{\min}$ at $h = 0.3$ and $t = H$.

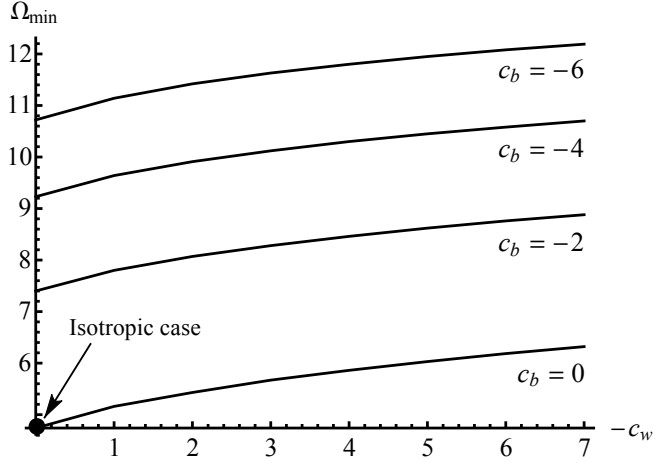

**Figure 4.** Effect of plastic anisotropy on $\Omega_{\min}$ at $h = 0.1$ and $t = 0$.

## 4. Limit Load of Cracked Specimens

### 4.1. Middle Cracked Specimen

This specimen is considered separately because of its extensive use in experiments [3,13,15]. The configuration of the specimen is shown in Figure 5.

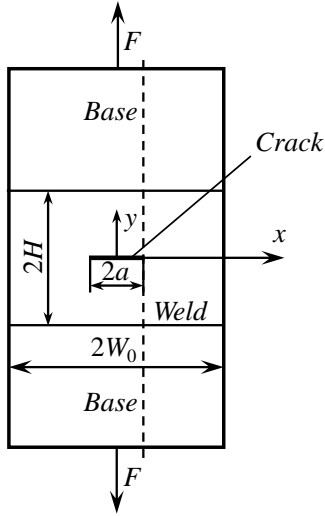

**Figure 5.** Middle cracked specimen.

The length of the crack is *2a*. The specimen is symmetric relative to the *y*-axis. Therefore, it is sufficient to consider the domain $x \geq 0$. The plastic work rate in the domain on the right to the dashed line (Figure 5) is equal to $\Omega_0$ from (22) if $W = W_0 - a$. The velocity discontinuity lines *db* and *de* (Figure 2b) should intersect at the crack tip. In this case, it is necessary to put $t = H$ in the solution given in Section 3. The plastic work rate vanishes in the region $0 \leq x \leq W_0 - a$. Therefore, it follows from the upper bound theorem that [7]

$$F_u U = \Omega_0 \tag{27}$$

where $F_u$ is an upper bound on the force *F*. Equations (23) and (27) combine to give

$$f_u = \frac{F_u}{W_0 B T_B} = \frac{W}{W_0} \Omega = \left(1 - \frac{a}{W_0}\right) \Omega. \tag{28}$$

Here, $f_u$ is the dimensionless representation of $F_u$. The best upper bound limit load based on the kinematically admissible velocity field adopted is obtained if $\Omega = \Omega_{\min}$. Using the solution of the auxiliary problem provided in Section 3, the upper bound limit load for the middle cracked specimen is immediate from (28) where $\Omega$ should be replaced with $\Omega_{\min}$. The solution illustrated in Figure 3 can be used for finding $\Omega$ involved in (28) if $H/(W_0 - a) = 0.3$.

### 4.2. Specimen Containing an Arbitrary Straight Crack inside the Weld

The configuration of this specimen is shown in Figure 6. The crack is straight and lies inside the weld, otherwise it is arbitrary. The location of the crack tips is completely determined by $W_p$, $W_l$, $t_p$, and $t_l$. It is assumed that these parameters are prescribed. Consider two domains separately. One of these domains, domain *p*, lies on the right to the right dashed line. The other domain, domain *l*, lies on the left to the left dashed line. The plastic work rate in domain *p* is equal to $\Omega_0$ from (23) if $W = W_p$. The velocity discontinuity lines *db* and *de* (Figure 2b) should intersect at the crack tip. In this case, it is necessary to put $t = t_p$ in the solution given in Section 3. This value of $\Omega_0$ will be denoted as $\Omega_0^{(p)}$ and the corresponding values of $\Omega$ and $\Omega_{\min}$ as $\Omega^{(p)}$ and $\Omega_{\min}^{(p)}$, respectively.

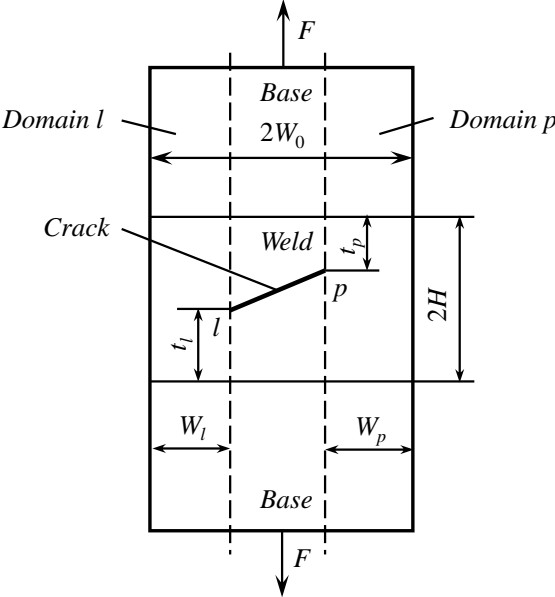

**Figure 6.** Geometry of the specimen containing an arbitrary straight crack.

The plastic work rate in domain $l$ is equal to $\Omega_0$ from (23) if $W = W_l$. The velocity discontinuity lines $db$ and $de$ (Figure 2b) should intersect at the crack tip. In this case, it is necessary to put $t = t_l$ in the solution given in Section 3. This value of $\Omega_0$ will be denoted as $\Omega_0^{(l)}$ and the corresponding values of $\Omega$ and $\Omega_{\min}$ as $\Omega^{(l)}$ and $\Omega_{\min}^{(l)}$, respectively.

The plastic work rate in the domain between the two dashed lines in Figure 6 vanishes. Therefore, it follows from the upper bound theorem that [7]

$$2F_u U = \Omega_0^{(p)} + \Omega_0^{(l)}. \tag{29}$$

Equations (23) and (29) combine to give

$$f_u = \frac{F_u}{W_0 B T_B} = \frac{W_p}{2W_0}\Omega^{(p)} + \frac{W_l}{2W_0}\Omega^{(l)}. \tag{30}$$

The values of $\Omega^{(p)}$ and $\Omega^{(l)}$ are independent of each other. Therefore, the best upper bound limit load based on the kinematically admissible velocity field adopted is obtained if $\Omega^{(p)} = \Omega_{\min}^{(p)}$ and $\Omega^{(l)} = \Omega_{\min}^{(l)}$. Using the solution of the auxiliary problem provided in Section 3, the upper bound limit load for the specimen under consideration is immediate from (30) where $\Omega^{(p)}$ and $\Omega^{(l)}$ should be replaced with $\Omega_{\min}^{(p)}$ and $\Omega_{\min}^{(l)}$, respectively.

The solution illustrated in Figures 3 and 4 can be used for several special cases. As an example, consider a specimen whose material parameters are the same as those adopted in Figures 3 and 4, and assume that $H/W_p = 0.1$, $H/W_l = 0.3$, $t_p = 0$ and $t_l = H$. Equation (30) can be rewritten as

$$f_u = \frac{W_p}{2W_0}\left[\Omega_{\min}^{(p)} + \frac{W_l}{W_p}\Omega_{\min}^{(l)}\right]. \tag{31}$$

Here, $\Omega_{\min}^{(l)}$ is determined from Figure 3 and $\Omega_{\min}^{(p)}$ from Figure 4. Moreover, $f_u$ is understood as the best upper bound limit load based on the kinematically admissible velocity field chosen. Using the geometric parameters prescribed Equation (31) becomes

$$f_u = \frac{W_p}{2W_0}\Lambda \tag{32}$$

where $\Lambda = \Omega_{min}^{(p)} + \Omega_{min}^{(l)}/3$. The solution (32) is valid for any value of $W_0$ satisfying the condition $W_l + W_p \leq 2W_0$. The variation of $\Lambda$ with $c_w$ at several values of $c_b$ is depicted in Figure 7.

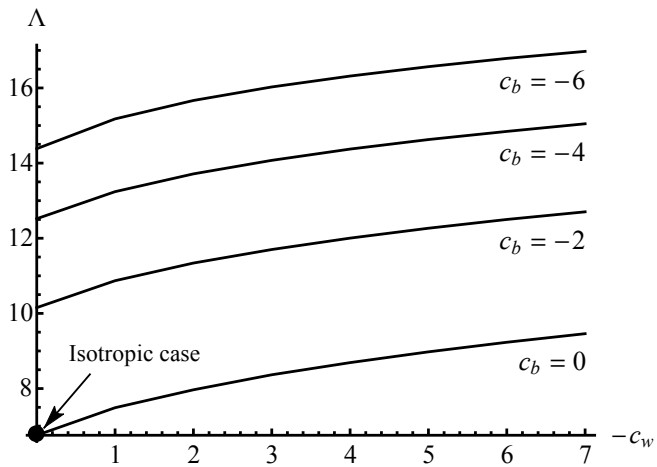

**Figure 7.** Effect of plastic anisotropy on the limit load.

### 4.3. Specimen with an Arbitrary Crack inside the Weld

The configuration of this specimen is shown in Figure 8. The crack lies inside the weld. The location of the crack tips is entirely determined by $W_p$, $W_l$, $t_p$, and $t_l$. It is assumed that these parameters are prescribed. The solution (30) is valid if the shape of the crack does not prevent the motion of rigid blocks 1 and 3 (Figure 2b) in opposite directions along the $y$-axis. This condition is satisfied if $-\pi/2 \leq \gamma \leq \pi/2$ (or $\pi/2 \leq \gamma \leq 3\pi/2$) everywhere. Here, $\gamma$ is the angle between the $y$-axis and the normal vector $\nu$ measured from the axis clockwise (Figure 8).

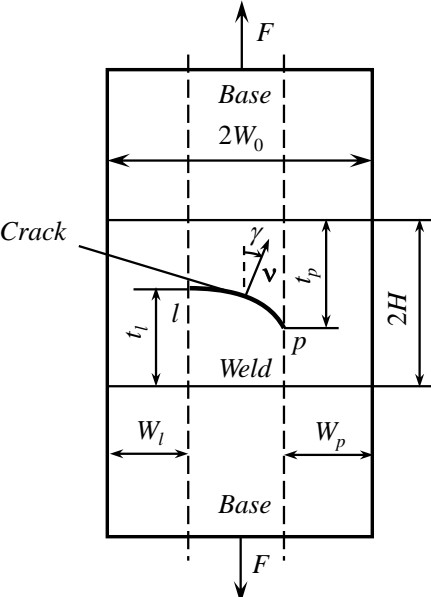

**Figure 8.** Geometry of the specimen containing an arbitrary curvilinear crack.

## 5. Conclusions

A new semi-analytic plane strain upper bound limit load solution for a welded tension specimen has been presented. The specimen contains a crack of arbitrary shape located inside the weld. Both the weld and base materials are orthotropic and obey Hill's quadratic yield criterion.

The solution is based on an efficient three-step procedure. Firstly, the solution of an auxiliary problem is found. Secondly, it is shown that a simple combination of two solutions of the auxiliary problem supplies an upper bound limit load for the welded tension specimen containing an arbitrary straight crack inside the weld. Finally, it is shown that this upper bound solution is valid for the specimen containing a curvilinear crack if the shape of the crack satisfies certain restrictions. The restrictions are derived.

The solution found reveals the significant effect of plastic anisotropy on the limit load (Figure 7). The magnitude of the limit load affects the accuracy of predictions based on many flaw assessment procedures. Therefore, the results obtained suggest that limit load solutions based on anisotropic plasticity should be used in applications of flaw assessment procedures for a wide class of materials. For practical applications, it is useful to have a compendium of limit load solutions for welded joints made of anisotropic materials similar to the existing compendium for the isotropic case [3].

The solution in the present paper is for the yield criterion proposed in [7]. However, it can be easily adapted for other pressure-independent yield criteria. To this end, it is only necessary to derive the shear stress on characteristic curves. This stress should replace the right-hand side of (20). A review of such anisotropic yield criteria is provided in [27], and it has been shown in [28] that the equations of plane strain deformation are hyperbolic for any criterion of this type (i.e., real characteristic curves exist).

**Author Contributions:** Conceptualization, E.L.; formal analysis, N.K.; writing, D.K.N. All authors have read and agreed to the published version of the manuscript.

**Funding:** This research was made possible by grants RFBR-18-51-54002 (Russia) and QTRU01.07/20-21 (Vietnam).

**Conflicts of Interest:** The authors declare no conflict of interest.

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
