# Peer review of "Influence of Plastic Anisotropy on the Limit Load of an Overmatched Cracked Tension Specimen"

_symmetry, doi:10.3390/sym12071079_

Round 1

Reviewer 1 Report

sents semi-analytical solutions for three-layered body containing crack. The solutions is obtained in the framework of the classical limit analysis theory. Novelty of the presented results is incremental and lies in the fact of application of the anisotropic Hill yield condition. However, a very special case is considered in which anisotropy axes for both materials are aligned and also agree with the loading direction. Do these conditions represent real situation or are only assumed for the simplification? This is not clear from the paper. Also, how much is it important that the Hill condition is quadratic and leads to the simplified flow rule as compared to non-linear yield conditions for the applicability of results?

Otherwise Reviewer does not see very important deficiencies of the paper but it also does not bring any breakthrough results. The paper can be accepted for publication, however, in spite of addressing the general issues indicated above, the following remarks should be taken into account:

1) formula 23-24 seems to contain an error

2) according to fig. 3,4 and 7, it seems that we may only increase the value of Omega_min as compared to isotropic case, independently on c and other parameters studied. However, not the whole scope of parameters is studied. For example, why admissible positive values of c in the regime (0,1) are not taken into account? Can we decrease the value of the limit load in such case?

3) In Fig 5 the dashed line, mentioned when the figure is referenced in the text, is not visible

4) I think that the section 4.3 can be merged with the previous one and the general crack can be considered from the start.

Author Response

The corrections made in the revised manuscript are shown in yellow.

Comment: The paper presents semi-analytical solutions for three-layered body containing crack. The solutions is obtained in the framework of the classical limit analysis theory. Novelty of the presented results is incremental and lies in the fact of application of the anisotropic Hill yield condition. However, a very special case is considered in which anisotropy axes for both materials are aligned and also agree with the loading direction. Do these conditions represent real situation or are only assumed for the simplification?

Answer: One of the most widely used processes for producing sheets is rolling. In this case, the principal axes of anisotropy are directly exactly as assumed in the manuscript. We have clarified this point in the revised manuscript. Any other orientation of the principal axes of anisotropy does not add any difficulty, as it follows from the upper bound theorem.

Comment:  Also, how much is it important that the Hill condition is quadratic and leads to the simplified flow rule as compared to non-linear yield conditions for the applicability of results?

Answer: We do not understand this comment. The yield criterion we use is non-linear. In any case, the plastic work rate is involved in the formulation of the upper bound theorem. Therefore, the particular yield criterion is not important for the general theory. We have clarified it in the revised manuscript.

Otherwise Reviewer does not see very important deficiencies of the paper but it also does not bring any breakthrough results. The paper can be accepted for publication, however, in spite of addressing the general issues indicated above, the following remarks should be taken into account:

Comment:

  • formula 23-24 seems to contain an error

Answer:

It is a misprint, indeed. Thank you so much. We have corrected it.

Comment:

  • according to fig. 3,4 and 7, it seems that we may only increase the value of Omega_min as compared to isotropic case, independently on c and other parameters studied. However, not the whole scope of parameters is studied. For example, why admissible positive values of c in the regime (0,1) are not taken into account? Can we decrease the value of the limit load in such case?

Answer:

The range of the c-value is -Infinity <c<1. There is nothing special about the range 0<c<1. We have to choose a finite interval of c on figures. The dependence of Omega_min on c_b at a given c_w (or on c_w at a given c_b) is monotonic. Therefore, for example, the limit load at c_b > 0 is smaller than that at c_b = 0. We think that it is evident from Figs. 3, 4, and 7.

Comment:

  • In Fig 5 the dashed line, mentioned when the figure is referenced in the text, is not visible

Answer:

Thank you. We have corrected it.

Comment:

  • I think that the section 4.3 can be merged with the previous one and the general crack can be considered from the start.

Answer:

It is, of course, possible, and it is easy to make this change. However, we think that the general line of reasoning is much more apparent if the curved crack is treated separately. Also, this line of reasoning can be used for other configurations.

Reviewer 2 Report

This paper studies the effect of plastic anisotropy on the strength of cracked specimen under tension. A fracture mechanics analysis is conducted, and a parametric study is performed to investigate the roles of different parameters. Overall this paper is well written, but the topic is old. There are a lot of studies on the topic many years ago. So, the novelty of this study needs to be clearly stated. The technical contribution of this research must be justified. 

I can see that you propose a new semi-analytic solution of the problem. However, the novelty and benefits of your solution are unclear. I am not sure about the significance of this study. 

In addition to the new solution that possibly has advantages, what are the other findings from this study? I saw that you show the effect of plastic anisotropy in Fig. 7. I think this is straightforward and insufficient. Please comment. 

Author Response

The corrections made in the revised manuscript are shown in yellow.

Comment: This paper studies the effect of plastic anisotropy on the strength of cracked specimen under tension. A fracture mechanics analysis is conducted, and a parametric study is performed to investigate the roles of different parameters. Overall this paper is well written, but the topic is old. There are a lot of studies on the topic many years ago. So, the novelty of this study needs to be clearly stated. The technical contribution of this research must be justified. 

Answer: Yes, of course, a lot of studies have been conducted on this topic. It shows that this topic is very important. In the literature, we have not found any solution for the problem we studied. We would really appreciate it if you let us know of any solution. Otherwise, the technical contribution is justified. Besides, we have included a review paper in the list of references to emphasize the importance of plastic anisotropy.

Comment: I can see that you propose a new semi-analytic solution of the problem. However, the novelty and benefits of your solution are unclear. I am not sure about the significance of this study. 

Answer: The novelty of the solution is that we take into account plastic anisotropy of both weld and base materials on the limit load. We are not aware of any other solution that quantifies this effect, and such solutions are needed for flaw assessment procedures.

Comment: In addition to the new solution that possibly has advantages, what are the other findings from this study? I saw that you show the effect of plastic anisotropy in Fig. 7. I think this is straightforward and insufficient. Please comment. 

Answer: It is, of course, straightforward to state that plastic anisotropy (as well as any other material property) affects the performance of structures. However, it is not so easy to quantify this effect. The solution provided in our paper allows one to find exact numbers that show the difference between isotropic and anisotropic cases. Moreover, we have proposed a method for taking into account the presence of various cracks in limit load solutions using a solution for the specimen with no crack. We think that this method is efficient because there are a lot of parameters that classify a welded structure with a crack. It is worthy of note that the method is not just the formal use of the upper bound theorem. For example, it would follow from this theorem that t = H in the solution given in Section 3.

Round 2

Reviewer 1 Report

Authors have addressed most of my comments.
However, the originality and impact od presented research are still incremental.

One comment still requires authors response, namely

Comment:

according to fig. 3,4 and 7, it seems that we may only increase the value of Omega_min as compared to isotropic case, independently on c and other parameters studied. However, not the whole scope of parameters is studied. For example, why admissible positive values of c in the regime (0,1) are not taken into account? Can we decrease the value of the limit load in such case?

Answer:

The range of the c-value is -Infinity <c<1. There is nothing special about the range 0<c<1. We have to choose a finite interval of c on figures. The dependence of Omega_min on c_b at a given c_w (or on c_w at a given c_b) is monotonic. Therefore, for example, the limit load at c_b > 0 is smaller than that at c_b = 0. We think that it is evident from Figs. 3, 4, and 7.

The answer is not satisfactory. I understand that the dependence is monotonic but if so, it would mean that for 1>c>0 we would observe a critical load which is smaller for anisotropic case than for isotropic one, which gives us qualitative difference in the impact of anisotropy on material/component performance - negative as compared to positive in the presented regime of c_b and c_w values. Could you comment on that issue?

Author Response

The corrections made in the revised manuscript are shown in yellow.

One comment still requires authors response, namely

Comment:

according to fig. 3,4 and 7, it seems that we may only increase the value of Omega_min as compared to isotropic case, independently on c and other parameters studied. However, not the whole scope of parameters is studied. For example, why admissible positive values of c in the regime (0,1) are not taken into account? Can we decrease the value of the limit load in such case?

Answer:

The range of the c-value is -Infinity <c<1. There is nothing special about the range 0<c<1. We have to choose a finite interval of c on figures. The dependence of Omega_min on c_b at a given c_w (or on c_w at a given c_b) is monotonic. Therefore, for example, the limit load at c_b > 0 is smaller than that at c_b = 0. We think that it is evident from Figs. 3, 4, and 7.

The answer is not satisfactory. I understand that the dependence is monotonic but if so, it would mean that for 1>c>0 we would observe a critical load which is smaller for anisotropic case than for isotropic one, which gives us qualitative difference in the impact of anisotropy on material/component performance - negative as compared to positive in the presented regime of c_b and c_w values. Could you comment on that issue?

Answer #2. You are right, but we have said it in the first answer (Therefore, for example, the limit load at c_b > 0 is smaller than that at c_b = 0.). Our manuscript does not say that the effect of anisotropy is positive. We have clarified it in the revised manuscript.

Also, it is important to mention that the comparison in question should be considered in conjunction with Eq.(1) at c = 0. This equation states that it is assumed in the paper that the shear yield stress of the isotropic material is equal to the shear yield stress of the anisotropic material in the xy – plane. However, it is possible to choose any other yield stress as the reference value. For example, assume that the tensile yield stress of the isotropic material is equal to X. Then, for comparison with the anisotropic case, it is necessary to multiply the dimensionless limit load for the isotropic material by X*Sqrt[3]/T. This number may be larger or smaller than 1. Therefore, the final result depends on our choice of the isotropic reference material. It is impossible to make this choice unambiguously. Therefore, we do not think that the effect of plastic anisotropy may be classified as positive or negative. We have clarified it in the revised manuscript (After Fig. 4).

Reviewer 2 Report

You did not well revise the manuscript based on review comments. My concerns were not addressed. 

A major concern is that you failed to clarify the novelty in this study. It is insufficient to say "Most of these solutions are upper bound solutions for isotropic materials". You must clearly state what is known and what is unknown.

The novelty is not obvious for this old topic. A lot of research on the effect of plastic anisotropy was conducted in 1960s to 1980s. Because many studies were already performed long time ago, there is less research in the past one to two decades. In the manuscript, you failed to point out the related studies. 

In fact, new advances have been made in the past one to two decades. Unfortunately, the progress is not introduced in the manuscript. The following papers might be useful to you:

Legarth, B.N., Tvergaard, V. & Kuroda, M. Effects of plastic anisotropy on crack-tip behaviour. International Journal of Fracture 117, 297–312 (2002). 

Benzerga, A.A., Besson, J. and Pineau, A., 2004. Anisotropic ductile fracture: Part II: theory. Acta Materialia52(15), pp.4639-4650.

Tanguy, B., Luu, T.T., Perrin, G., Pineau, A. and Besson, J., 2008. Plastic and damage behaviour of a high strength X100 pipeline steel: Experiments and modelling. International Journal of Pressure Vessels and Piping85(5), pp.322-335.

Frodal, B.H., Morin, D., Børvik, T. and Hopperstad, O.S., 2020. On the effect of plastic anisotropy, strength and work hardening on the tensile ductility of aluminium alloys. International Journal of Solids and Structures188, pp.118-132.

Author Response

The corrections made in the revised manuscript are shown in yellow.

Comment 1: A major concern is that you failed to clarify the novelty in this study. It is insufficient to say "Most of these solutions are upper bound solutions for isotropic materials". You must clearly state what is known and what is unknown.

Answer: We thought that it was sufficient to refer to review papers. In the revised manuscript, we provided our short review of limit load solutions for anisotropic materials. It is seen from this review that our solution is new.

Comment 2: The novelty is not obvious for this old topic. A lot of research on the effect of plastic anisotropy was conducted in 1960s to 1980s. Because many studies were already performed long time ago, there is less research in the past one to two decades. In the manuscript, you failed to point out the related studies. 

In fact, new advances have been made in the past one to two decades. Unfortunately, the progress is not introduced in the manuscript. The following papers might be useful to you:

Legarth, B.N., Tvergaard, V. & Kuroda, M. Effects of plastic anisotropy on crack-tip behaviour. International Journal of Fracture 117, 297–312 (2002). 

Benzerga, A.A., Besson, J. and Pineau, A., 2004. Anisotropic ductile fracture: Part II: theory. Acta Materialia52(15), pp.4639-4650.

Tanguy, B., Luu, T.T., Perrin, G., Pineau, A. and Besson, J., 2008. Plastic and damage behaviour of a high strength X100 pipeline steel: Experiments and modelling. International Journal of Pressure Vessels and Piping85(5), pp.322-335.

Frodal, B.H., Morin, D., Børvik, T. and Hopperstad, O.S., 2020. On the effect of plastic anisotropy, strength and work hardening on the tensile ductility of aluminium alloys. International Journal of Solids and Structures188, pp.118-132.

Answer: Effect of plastic anisotropy on what? Our paper deals with the limit load considered as an input parameter of the defect assessment procedures. The only important material property for calculating the limit load is the yield criterion, assuming that the associated flow rule is valid. All other studies on plastic anisotropy are not directly related to limit load solutions (not to our solution but all limit load solutions).

We referred to the papers you recommended. However, the last three papers on your list are not related to the objective of our paper at all. These papers deal with FE modeling of plastic deformation and damage. Of course, plastic anisotropy is taken into account, but without any contribution to limit load solutions (according to the classical definition of the limit load).

Please note that we also added papers devoted to limit load solutions. These papers were published at the same time as the papers you suggested (except the last paper on your list). Our paper contributes to this area of research.

Round 3

Reviewer 2 Report

I agree this paper reports something new, but the contribution is incremental and insufficient. I do not have other comments.